# Blurring the boundary between homogenous and heterogeneous catalysis using palladium nanoclusters with dynamic surfaces

Israel Cano[1], Andreas Weilhard[1], Carmen Martin[1], Jose Pinto[1,2], Rhys W. Lodge[1], Ana R. Santos[2], Graham A. Rance[1,3], Elina Harriet Åhlgren[1], Erlendur Jónsson[4,5], Jun Yuan[6], Ziyou Y. Li[7], Peter Licence[2], Andrei N. Khlobystov[1] & Jesum Alves Fernandes[1✉]

Using a magnetron sputtering approach that allows size-controlled formation of nanoclusters, we have created palladium nanoclusters that combine the features of both heterogeneous and homogeneous catalysts. Here we report the atomic structures and electronic environments of a series of metal nanoclusters in ionic liquids at different stages of formation, leading to the discovery of Pd nanoclusters with a core of *ca.* 2 nm surrounded by a diffuse dynamic shell of atoms in $[C_4C_1Im][NTf_2]$. Comparison of the catalytic activity of Pd nanoclusters in alkene cyclopropanation reveals that the atomically dynamic surface is critically important, increasing the activity by a factor of *ca.* 2 when compared to compact nanoclusters of similar size. Catalyst poisoning tests using mercury and dibenzo[*a,e*] cyclooctene show that dynamic Pd nanoclusters maintain their catalytic activity, which demonstrate their combined features of homogeneous and heterogeneous catalysts within the same material. Additionally, kinetic studies of cyclopropanation of alkenes mediated by the dynamic Pd nanoclusters reveal an observed catalyst order of 1, underpinning the pseudo-homogeneous character of the dynamic Pd nanoclusters.

[1] School of Chemistry, University of Nottingham, Nottingham, UK. [2] GSK Carbon Neutral Laboratories for Sustainable Chemistry, University of Nottingham, Nottingham, UK. [3] Nanoscale and Microscale Research Centre, University of Nottingham, Nottingham, UK. [4] Department of Chemistry, University of Cambridge, Cambridge, UK. [5] Department of Physics, Chalmers University of Technology, Gothenburg, Sweden. [6] Department of Physics, University of York, York, UK. [7] Nanoscale Physics Research Laboratory, School of Physics and Astronomy, University of Birmingham, Birmingham, UK. ✉email: jesum.alvesfernandes@nottingham.ac.uk

Catalysts are used in nearly 80 % of industrial processes[1], demanding the maximum catalytic efficiency of rare elements, such as Au, Pt and Pd[2–4]. Increasing the metal active surface area by utilising catalysts in the form of nanoclusters is one of the most powerful approaches[5–9]. When the size of metal shrinks down to the nanoscale, the structure and dynamics and thus properties of the nanocluster cannot be interpreted as a linear function of its size, which makes it challenging to correlate them with their catalytic performance[7,10–12]. In order to establish a structure–property relationship for nanoclusters and the link with their catalytic performance, we must explore the series of related nanoclusters at their boundaries between discrete atoms and crystalline nanoclusters, with atomic precision, and using a set of complementary analytical methods, thus shedding light on their nature[13].

In this context, magnetron sputtering offers exciting new opportunities for the manufacture of novel catalytic materials with controlled sizes and shapes of metal nanoclusters on a variety of supports[14–18]. Crucially, this approach generates highly active clean surfaces, since no chemical stabilisers and/or solvents are involved in the process[16,19]. Among the supports, ionic liquids (ILs) are unique because they offer a soft, liquid environment for relatively free (in comparison to solid supports) nanoclusters to be studied in their native state. Furthermore, ILs possess low vapour pressure allowing their use under the high vacuum conditions of magnetron sputtering[20,21]. However, atomic-scale analysis of metal nanoclusters in IL, for example, by transmission electron microscopy (TEM), remains a challenge. The current sample preparation approaches using stabilisers or solvents[22–24] may alter the environment surrounding the nanocluster, thus potentially inducing structural changes to the nanocluster and therefore hide important information on its shape and size. To the best of our knowledge, metal nanoclusters in ILs deposited by magnetron sputtering have not been observed with atomic resolution in their native state so far. As a result, the relationship between metal nanoclusters structure and their catalytic activity in ILs remains largely underexplored and widely debated.

Herein we determine the atomic structures and electronic environments of a series of metal nanoclusters in ILs at different stages of formation, leading to a discovery of Pd nanoclusters with a core of ca. 2 nm surrounded by a diffuse dynamic shell of atoms in 1-butyl-3-methylimidazolium bis(trifluoromethylsulfonyl)imide ($[C_4C_1Im][NTf_2]$) that combines the features of homogeneous and heterogeneous catalysts within the same material.

## Results and discussion

**Synthesis of Pd nanoclusters.** The magnetron sputtering metal deposition process involves the elastic collision of argon ions with a highly pure metallic target, resulting in the expulsion of atoms or clusters from the target that are deposited onto a support material (Fig. 1a and Supplementary Fig. 1)[14]. Previously, it was proposed that the growth of metal clusters occurs on the IL surface/near surface or in the bulk IL depending on the magnetron sputtering conditions and physical properties of the employed IL[25–27]. However, a variety of parameters, including the applied potential, argon (Ar) working pressure and work distance (distance between the metal target and the support), can influence: (i) whether metal atoms or a few metal atom cluster are ejected from the target; (ii) the possible formation of metal clusters in the gas phase; (iii) the kinetic energy of the metal atoms/clusters landing on the IL and thus how deeply they will penetrate. Additionally, the physical properties of the IL (i.e. surface tension) or substituent functional groups also play a major role in the metal cluster growth[20].

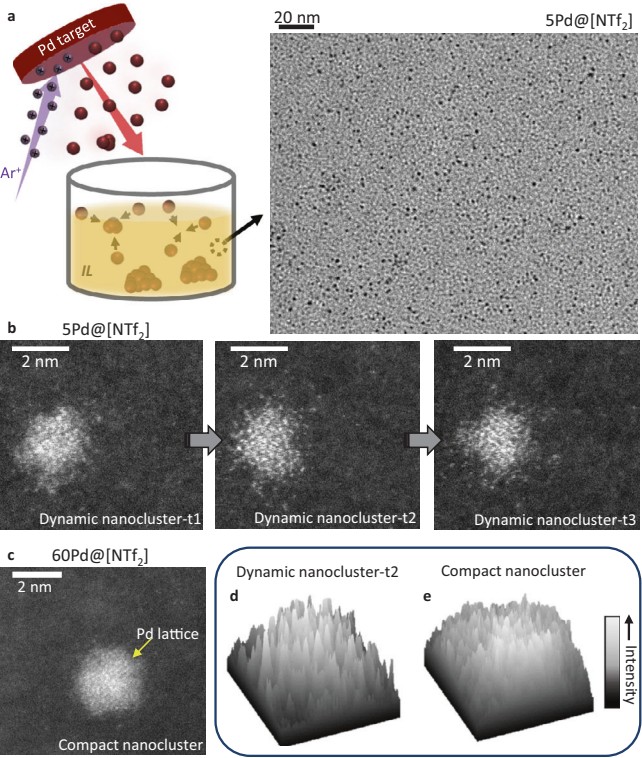

**Fig. 1 The magnetron sputtering process and electron microscopy images of Pd nanoclusters in ILs. a** Scheme of the Pd magnetron sputtering deposition in IL. Bright-field TEM image shows the homogenous size distribution of 5Pd@[$NTf_2$]. **b** Aberration-corrected scanning transmission electron microscopic (AC-STEM) images acquired over time (details in Supplementary Fig. 8) to show the dynamic character of 5Pd@[$NTf_2$] system. **c** AC-STEM image of Pd species deposited for 60Pd@[$NTf_2$] (details in Supplementary Fig. 11) shows nanocluster with a more compact structure compared to 5Pd@[$NTf_2$], with metallic Pd lattice planes being observed. **d, e** Intensity histograms were taken from AC-STEM images to highlight the difference between compact and dynamic Pd nanoclusters surfaces, respectively. The intensity bar refers to STEM image contrast, which correlates with the number of Pd atoms in the image.

In this work, magnetron sputtering depositions of Pd species into ILs were carried out using ultraclean conditions with a background-pressure of $5.3 \times 10^{-6}$ Pa, work pressure fixed at $4.0 \times 10^{-1}$ Pa using Ar gas (99.9995%) and a high purity Pd target (99.995%). Theoretical simulations for different potentials were performed prior to Pd depositions in IL to optimise the ejection of single Pd atoms (see Supplementary Methods for further details). The simulations showed the highest probability for single Pd atoms to be sputtered from the target at applied potentials up to 800 eV, with two- or three-atom Pd clusters being minority species (i.e. at 320 eV, 31 single Pd atoms are ejected for each two-atom cluster—Supplementary Fig. 2). Therefore, the Pd depositions in ILs were performed using a potential applied within the range displayed in Supplementary Fig. 2. The only parameters varied during the depositions were the Pd concentration and the nature of IL anion. Further details of the magnetron sputtering parameters can be found in Supplementary Table 1 and Supplementary Methods. As expected, the Pd concentration in ILs increased linearly with the Pd deposition time (Supplementary Fig. 3). Therefore, the resulting materials were named as XPd@Y where X is Pd deposition time in min and Y is the nature of the anion in [$C_4C_1Im$]. Y as the IL cation was kept constant. For instance, Pd species deposited for 5 min (Pd concentration of 0.12 wt%) in [$C_4C_1Im$][$NTf_2$] would be denoted as 5Pd@[$NTf_2$].

**Pd nanocluster morphology and atomic structure**. All TEM analyses were performed directly in IL to evaluate the true morphology and atomic structure of the Pd species. Bright-field TEM imaging revealed variation of Pd nanocluster size distribution, from sub-1 nm in 1Pd@[NTf$_2$] to 2–3 nm in all other ILs (Supplementary Figs. 4, 5, 9, 10 and 12).

Atomic-scale investigation using aberration-corrected scanning transmission electron microscopy (AC-STEM) for 5Pd@[NTf$_2$] revealed unusual features—a Pd core surrounded by a diffuse shell of satellite Pd atoms, with the overall nanocluster diameter of 2 ± 1.0 nm (Fig. 1b). The structure is dynamic with individual Pd atoms reshuffling over time, as revealed by time-series AC-STEM imaging, and henceforth is referred to as a dynamic nanocluster (Fig. 1b and Supplementary Figs. 6–8). However, the further increase of the Pd concentration in [C$_4$C$_1$Im][NTf$_2$] (Fig. 1c and Supplementary Fig. 11), or the use of a different IL with the anion [PF$_6$]$^-$ (Supplementary Fig. 13), produces a compact Pd nanocluster with significantly lower dynamics than 5Pd@[NTf$_2$]. In the compact Pd nanoclusters, atomic lattice planes (111) of face-centred cubic Pd are observed (Supplementary Figs. 9, 10 and 12), whereas in 5Pd@[NTf$_2$] it appears to be amorphous (Fig. 1b and Supplementary Fig. 5). The overall three-dimensional disorder of the dynamic Pd nanoclusters is also apparent in the irregular intensity distribution extracted from AC-STEM images (Fig. 1d and Supplementary Fig. 8b), as compared to a more homogenous intensity pattern typical of the compact nanoclusters such as 60Pd@[NTf$_2$] and 5Pd@[PF$_6$] (Fig. 1e and Supplementary Figs. 11b and 13b).

Matrix-assisted laser desorption ionisation time of flight mass spectrometry clearly shows the presence of single Pd atoms in Pd@[NTf$_2$] (Supplementary Figs. 14–20). The presence of single Pd atoms appears to be strongly dependent on the type of IL, as changing the IL cation ([C$_4$C$_1$Im]$^+$ to 1,2-dimethyl-3-butyl-imidazolium cation ([C$_4$C$_1$C$_1$Im]$^+$) or anion ([NTf$_2$]$^-$) to hexafluorophosphate anion ([PF$_6$]$^-$)) both have an impact on the availability of Pd atoms, such that no single Pd atoms are detected in [C$_4$C$_1$C$_1$Im][NTf$_2$] and [C$_4$C$_1$Im][PF$_6$]. As the loading of the metal increases, the proportion of Pd existing in the atomic form decreases, consistent with the AC-STEM observations that dynamic nanoclusters exist only in [C$_4$C$_1$Im][NTf$_2$] and only at low loading of metal.

**Pd nanocluster electronic environment**. A combination of X-ray photoelectron spectroscopy (XPS), X-ray absorption spectroscopy (XAS) and Raman spectroscopy provided insight into the interactions between Pd and [C$_4$C$_1$Im][NTf$_2$]. The Pd 3d core XPS spectrum indicates that Pd nanoclusters in [C$_4$C$_1$Im][NTf$_2$] exist in two different electronic environments: the metallic Pd peak at 335.9 eV and a peak at 338.5 eV that we assign to the interaction between surface Pd atoms with [C$_4$C$_1$Im][NTf$_2$] (Fig. 2a, Supplementary Figs. 21–25 and Supplementary Table 2). The integral intensity of metallic Pd$^0$ peak shows a relative decrease as the loading of Pd in [C$_4$C$_1$Im][NTf$_2$] decreases (from 60Pd@[NTf$_2$] to 10Pd@[NTf$_2$]) (Fig. 2a). These results correlate with the observation in AC-STEM and mass spectrometry, suggesting that surface Pd atoms interact with IL and that dynamic Pd nanoclusters exist mainly at low loadings.

X-ray absorption near-edge structure (XANES) measurements for 5Pd@[NTf$_2$] and 30Pd@[NTf$_2$] Pd K-edge show a similar adsorption edge ($E_0$) to bulk Pd metal but with a higher white line ($H_w$) intensity in the 30Pd@[NTf$_2$] and a further $H_w$ intensity increases for 5Pd@[NTf$_2$]. This indicates d-electron depletion of Pd nanoclusters due to charge transfer to the more electronegative groups of the [C$_4$C$_1$Im][NTf$_2$] IL (Fig. 2b)[28], consistent with the XPS results. Extended X-ray absorption fine structure (EXAFS) analysis of 5Pd@[NTf$_2$] and 30Pd@[NTf$_2$] show a peak at ca. 2.5 Å associated with Pd–Pd bond, same as in bulk metal, as well as another peak at ca. 1.7 Å that we assigned to Pd atom interaction with [C$_4$C$_1$Im][NTf$_2$] (Fig. 2c). Density functional theory (DFT) modelling confirmed that the Pd coordination

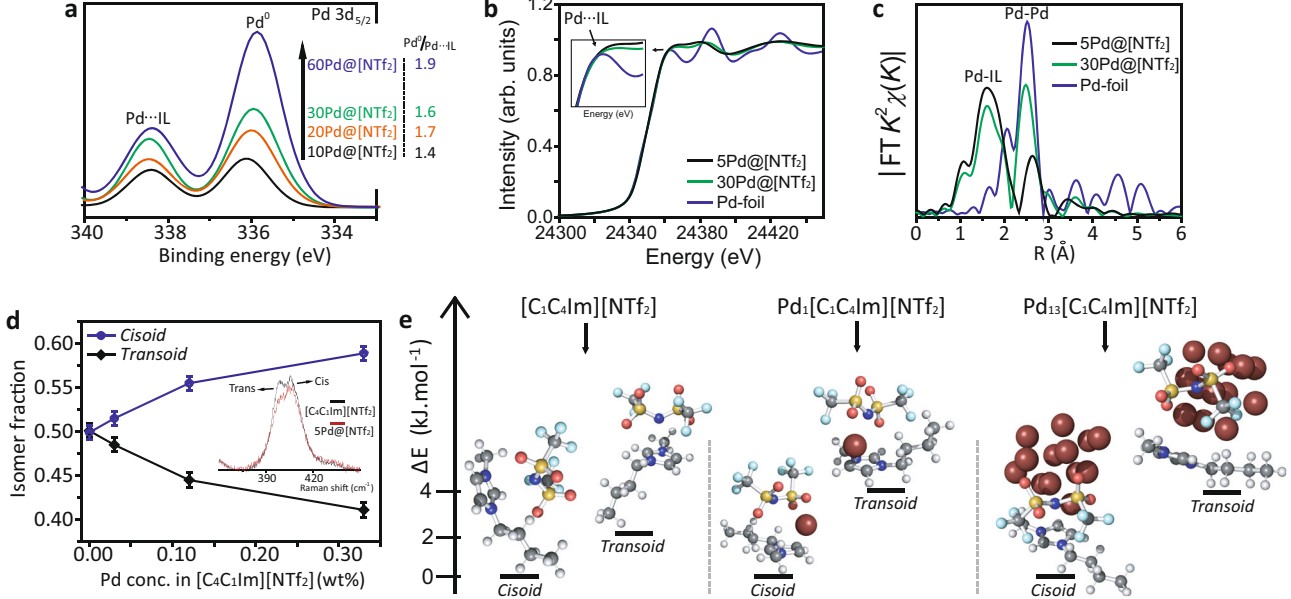

**Fig. 2 Electronic environment of Pd and ILs system. a** X-ray photoelectron spectroscopy (XPS) spectra for the Pd 3d$_{5/2}$ of 10Pd@[NTf$_2$], 20Pd@[NTf$_2$], 30Pd@[NTf$_2$] and 60Pd@[NTf$_2$] (no Pd signal was observed at lower Pd concentration—Supplementary Fig. 22a). **b, c** Pd K-edge X-ray absorption near-edge structure (XANES) and Fourier transform of k$^2$-weighted extended X-ray absorption fine structure (EXAFS) spectra, respectively, of 5Pd@[NTf$_2$], 30Pd@[NTf$_2$] and Pd foil (PdO spectrum can be found in Supplementary Fig. 26). Pd-IL: coordination of Pd-C from [C$_4$C$_1$Im] cation and/or Pd-O/N from [NTf$_2$] anion. **d** Relative cis-/trans- conformational isomer (conformer) fractions as a function of Pd concentration obtained from Raman spectroscopic analysis of pure [C$_4$C$_1$Im][NTf$_2$] and 1Pd@[NTf$_2$], 5Pd@[NTf$_2$] and 10Pd@[NTf$_2$]. **e** Energy difference between the cis- and trans- conformers for [C$_4$C$_1$Im][NTf$_2$], Pd$_1$[C$_4$C$_1$Im][NTf$_2$] and Pd$_{13}$[C$_4$C$_1$Im][NTf$_2$] from left to right, respectively.

occurs via N and O atoms of the $[NTf_2]^-$ anion for the $Pd_{13}[C_4C_1Im][NTf_2]$, whereas for the $Pd_1[C_4C_1Im][NTf_2]$ complex the coordination occurs via C-atoms and O-atoms from the cation and anion, respectively (Fig. 2e). Moreover, the intensity and the peak corresponding to Pd–Pd distance decreases as the Pd loading decreases ($30Pd@[NTf_2]$ to $5Pd@[NTf_2]$) alongside to a relative increase of the Pd-IL peak intensity, suggesting a higher number of Pd atoms interacting with the $[C_4C_1Im]$ cation and/or $[NTf_2]$ anion at lower Pd loading in IL, in agreement with the XPS analysis. Additionally, for $5Pd@[NTf_2]$ EXAFS indicated slightly longer Pd–Pd bonds compared to $30Pd@[NTf_2]$ and Pd bulk, which can be related to the dynamic Pd nanocluster behaviour at relatively low Pd loading in IL ($5Pd@[NTf_2]$)[29]. Both XPS and XAS demonstrate a strong interaction between surface Pd atoms and $[C_4C_1Im][NTf_2]$ and with the core Pd atoms remaining in metallic-like state. Raman spectroscopy allows probing the IL and shows the increase of *cis*- conformation of the $[NTf_2]$ anion, as the loading of Pd increases (Fig. 2d, Supplementary Fig. 27 and Supplementary Table 3). DFT modelling also shows that for $[C_4C_1Im][NTf_2]$ the *cis*- conformer is slightly more stable than the *trans*- conformer by 2.1 kJ mol$^{-1}$, the addition of a Pd atom, as in $Pd_1[C_4C_1Im][NTf_2]$ and $Pd_{13}[C_4C_1Im][NTf_2]$, increases the difference in stability of the conformers to 4.1 and 3.9 kJ mol$^{-1}$, respectively (Fig. 2e and Supplementary Fig. 28).

**Catalytic evaluation of Pd nanocluster.** The catalytic performance of the $Pd@[NTf_2]$ system was evaluated in a cyclopropanation of alkenes where Pd facilitates carbene transfer from ethyl diazoacetate (EDA) to alkenes (Fig. 3a, b, for further details, see Supplementary Tables 4 and 5 and Supplementary Notes 1)[30–33].

All materials were found to be active in the cyclopropanation of styrene, giving good conversions and selectivities towards the cyclopropane product. In order to compare relative catalytic conversions of different Pd nanoclusters, a series of experiments in $Pd@[NTf_2]$ with a normalised metal loading in the reaction mixture were carried out (0.5 mol% Pd) (Fig. 3b; for further details see, Supplementary Table 4 and Supplementary Notes 1). While selectivities were similar, a gradual increase in the conversion was observed with a decrease in the Pd metal concentration in IL (from $60Pd@[NTf_2]$ to $10Pd@[NTf_2]$), with an abrupt increase for $7Pd@[NTf_2]$, $5Pd@[NTf_2]$ and $1Pd@[NTf_2]$ (Fig. 3b and Supplementary Table 4), correlating with the transition from compact to dynamic Pd nanoclusters, indicating a higher fraction of accessible Pd atoms in the latter.

In order to verify the mode of catalysis by dynamic Pd nanoclusters, mercury poisoning tests were performed (Fig. 3c). It is well established that heterogeneous catalysts, in contrast to homogenous catalysts, lose their catalytic activity in the presence of Hg due to the poisoning of their surface[34]. Indeed, complete inhibition of the catalysis was observed for compact Pd nanoclusters (from $60Pd@[NTf_2]$ to $10Pd@[NTf_2]$; Fig. 3c), suggesting the absence and/or very low population of dynamic Pd nanoclusters in agreement with the AC-STEM observations. Remarkably, for the dynamic Pd nanoclusters, the catalytic activity was maintained for $1Pd@[NTf_2]$ and only 32% decrease in yield was observed for $5Pd@[NTf_2]$ (Fig. 3c and Supplementary Table 6), suggesting a pseudo-homogeneous mode of catalysis by the dynamic nanoclusters. Additional poisoning tests were conducted for higher Pd concentration in $[C_4C_1Im][NTf_2]$ and $5Pd@[PF_6]$ (Supplementary Table 6), which were completely deactivated similarly to traditional heterogeneous catalysts.

We have performed an additional poisoning tests using dibenzo[*a*,*e*]cyclooctene (DCT), which selectively binds to single Pd sites[34–37]. DCT experiments showed that $5Pd@NTf_2$ and $30Pd@NTf_2$ did not lose activity (Supplementary Table 7),

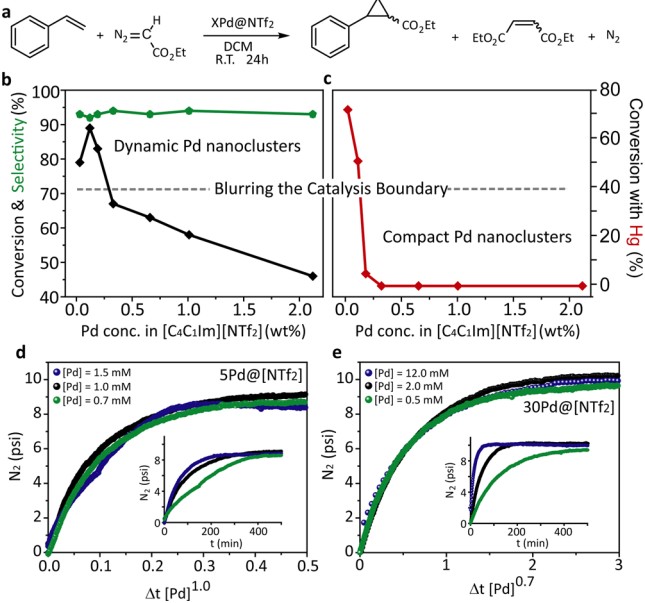

**Fig. 3 Cyclopropanation of styrene. a** Reaction scheme of the cyclopropanation of styrene as a model reaction with *X*Pd@*Y*. A complete substrate scope can be found in Supplementary Notes 1. **b** Conversion (black) and selectivity (green) with varying Pd contents in $[C_4C_1Im][NTf_2]$ determined after 24 h; reaction conditions: diazoethyl acetate (1 mmol), styrene (5 mmol), *X*Pd@$[NTf_2]$ (5 mol%, based on Pd) and DCM (5 mL) at room temperature. **c** Conversion with varying Pd contents in $[C_4C_1Im]$ $[NTf_2]$ and the presence of Hg, reaction conditions as in **b** with Hg. **b**, **c** Dashed line represents the boundary in between pseudo-homogeneous and classical heterogeneous catalysts. **d**, **e** represent the time normalised analysis of the $N_2$ release (which is coupled to the overall catalytic transformation) with varying catalyst concentrations. The catalyst order is represented as the magnitude of the [Pd] concentration and the original data can be found as insets in **d** and **e**, respectively. A catalyst order of 1 is found for $5Pd@[NTf_2]$ (**d**) and an order of 0.7 is found for $30Pd@[NTf_2]$ (**e**). Reaction conditions in **d**, **e**: 1 mmol ethyl diazoacetate, 5 mmol styrene, 5 mL DCM at 35 °C with varying concentrations of Pd but constant concentration of $[C_4C_1Im][NTf_2]$.

confirming that dynamic Pd nanocluster are the active sites, rather than leached single Pd atoms. These results were further reinforced by filtration tests, as both catalytic systems ($5Pd@NTf_2$ and $30Pd@NTf_2$) did not show activity after the filtration (Supplementary Table 8). Additionally, recyclability experiments showed that $5Pd@NTf_2$ and $30Pd@NTf_2$ are recyclable and maintain their activity over three cycles (Supplementary Table 9).

Furthermore, the $N_2$ release was monitored during the course of the reaction to elucidate the catalyst order (Fig. 3d, e and Supplementary Fig. 29). After pre-activation with styrene, the catalyst order was determined by the variable time normalisation analysis method introduced by Bures and co-workers[38–40], showing a catalyst order of 1.0 for $5Pd@[NTf_2]$ and 0.7 for $30Pd@[NTf_2]$ (Fig. 3d, e). These results further confirm the pseudo-homogeneous character of the dynamic Pd nanoclusters, while the order determined for compact Pd nanoclusters is in the expected lower range for heterogeneous catalysts due to the inherently decreased amounts of atoms accessible for catalysis. This effect relates to the condition that the reaction takes place only on the surface of the catalysts. TEM experiment where dynamic Pd nanoclusters were retrieved from the reaction mixture directly onto a TEM grid demonstrated that the dynamic clusters remain unchanged in size and structure after the reaction (Supplementary Fig. 30).

In summary, we have explored the structure, dynamics and catalytic properties of a series of palladium nanoclusters from sub-1 to 3 nm sizes by a variety of analytical and imaging techniques, providing atomic-scale information. Nanoclusters appear to behave as dynamic structures with properties controlled by the nature of IL (both cation and anion) and the loading of the metal in the liquid. Anions interacting directly with palladium atoms play a particularly important role in determining the electronic state and dynamics of surface atoms. In particular, $[NTf_2]$ anions in $[C_4C_1Im][NTf_2]$ have been shown to bond to surface Pd atoms and create a diffuse shell of metal atoms around the nanocluster. This phenomenon has a significant implication for the mode of catalysis, changing the heterogeneous behaviour, exhibited by Pd nanoclusters with compact structures, to a homogeneous mode exhibited by dynamic $5Pd@[NTf_2]$. Our study demonstrates that the effectiveness of nanocatalysts depends not only on the surface area of the nanoclusters but also on the dynamic behaviour of their surface atoms controlled by the support environment.

## Methods

**Pd nanocluster preparation**. Pd species were deposited into ILs (IL synthesis details are in the Supplementary Information) using a bespoke AJA magnetron sputtering system with a load-lock sample transfer facility coupled to a glovebox. The Pd target (99.995%) was purchased from AJA International. IL samples were loaded through a glovebox into the magnetron sputtering system to avoid the presence of moisture. In a typical experiment, anhydrous IL (0.750 g) is placed in a petri dish while in the glovebox. The sample was transferred to a load-lock under a $N_2$ environment, pumped-down to a background pressure of $5.3 \times 10^{-5}$ Pa for (ca. 1 h) and then transferred to the main chamber, which reached a background pressure of $5.3 \times 10^{-6}$ Pa in ca. 1 h. The power and working pressure used in all depositions were 100 W and $4.0 \times 10^{-1}$ Pa (Argon—99.9999%), respectively.

**Sample characterisation**. Pd nanoclusters deposited by magnetron sputtering in ILs were primarily characterised by TEM, with each being analysed from at least two different deposition batches. All microscopic analyses were performed in the absence of solvents in order to evaluate the true morphology of the Pd species deposited in the ILs. TEM measurements were performed using a JEOL 2100F FEG-TEM operated with an accelerating voltage of 200 kV. AC-STEM measurements were performed using a JEOL 2100F scanning transmission electron microscope with a CEOS aberration corrector operated with an accelerating voltage of 200 kV. The samples were prepared in the following manner: copper mesh, holey carbon film TEM grids (Agar Scientific, UK) were glow discharged (Agar Turbo Coater, 0.2 mbar, 5 mA, 10 s) before the addition of Pd@IL (10 μL). The Pd@IL suspension was kept on the grid for 2 min before the excess was removed using filter paper. This approach allowed the formation of 'pools' of IL within the holes of the holey carbon, providing an effective contrast-free region to observe the palladium structures.

XPS measurements were performed using a Kratos AXIS Ultra DLD instrument. The chamber pressure during the measurements was $6.7 \times 10^{-7}$ Pa. Wide energy range survey scans were collected at pass energy of 80 eV in hybrid slot lens mode and a step size of 0.5 eV, for 20 min. High-resolution data on the Pd 3d, C 1s, N 1s, O 1s, S 2p and F 1s photoelectron peaks were collected at a pass energy of 20 eV over energy ranges suitable for each peak and collection times of 5 min, step sizes of 0.1 eV. The charge neutraliser filament was used to prevent the sample charging over the irradiated area. The X-ray source was a monochromated Al Kα emission, run at 10 mA and 12 kV (120 W). The energy range for each 'pass energy' was calibrated using the Kratos Cu $2p_{3/2}$, Ag $3d_{5/2}$ and Au $4f_{7/2}$ three-point calibration method. The transmission function was calibrated using a clean gold sample method for all lens modes and the Kratos transmission generator software within Vision II. The data were processed with CASAXPS (Version 2.3.17). The high-resolution data were charge corrected to the reference F 1s signal at 688.9 eV.

XAS measurements of Pd K-edge were performed at room temperature (B18 beamline) at the Diamond Synchrotron Light. XANES and EXAFS spectra of a Pd foil and PdO standards were measured and the energy calibrated by aligning the respective absorption edges. The data were calibrated and normalised by a linear pre-edge subtraction using the ATHENA software. DFT calculations were performed using version 4.1.0 of the Orca Program package[41].

Micro Raman spectroscopy was performed using a Horiba Jobin Yvon LabRAM HR Raman spectrometer. Spectra were acquired using a 785 nm laser (at 24 mW power), a ×100 objective lens and a 300 μm confocal pinhole. To simultaneously scan a range of Raman shifts and control the spectral resolution, either a 600 or 1800 lines $mm^{-1}$ rotatable diffraction grating along a path length of 800 mm was employed. Spectra were acquired using a Synapse CCD detector (1024 pixels) thermoelectrically cooled to −60 °C.

**Catalytic tests**. All reactions were prepared in a glovebox under inert atmosphere using a 50 mL Schlenk flask equipped with a magnetic stir bar. EDA (1 mmol) was added into a solution containing the Pd catalyst (0.5 mol% Pd) and styrene (5 mmol) in $CH_2Cl_2$ (5/10 mL). The Schlenk flask was sealed with a septum cap and the reaction mixture was removed from the glovebox. After 24 h of stirring at RT, the mixture was filtered through silica gel and volatiles were removed in vacuo. Products have been previously described and their identification was straightforward from comparison with the reported data[42]. Substrate conversion and selectivity and yield of cyclopropane product were determined by $^1$H nuclear magnetic resonance spectroscopy using 1,2-dibromoethane (0.25 mmol) as an internal standard.

## Data availability

All experimental and simulation data in the main text and Supplementary Materials are available upon request to the authors.

## Code availability

Molecular Dynamics codes used in the simulation are available upon request to harriet. ahlgren@univie.ac.at.

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

## Acknowledgements

The authors thank the University of Nottingham Advanced Molecular Materials Interdisciplinary Research Cluster, Propulsion Futures and Green Chemicals Beacons of Excellence for the financial support, the Nanoscale and Microscale Research Centre (nmRC) for access to materials characterisation equipment and the National Mass Spectrometry Facility at Swansea University and Mark F. Wyatt. The authors acknowledge support of EPSRC CDT in Sustainable Chemistry (EP/L015633/1), EPSRC projects (EP/K005138/1) and BBSRC (BB/L013940/1) for the financial support. E.H.Å. thanks the funding from Eemil Aaltonen Foundation and the access to the UoN's Augusta HPC service. The authors acknowledge the Diamond Light Source and the UK Catalysis Hub for provision of beam time (proposal number SP19850-5 and through the Block Allocation Group (BAG) for Energy Materials under proposal sp17198) and Alan Chadwick and Giannantonio Cibin for assistance with the XAS measurements. UK Catalysis Hub provided resources and support funded by EPSRC grants: EP/R026939/1, EP/R026815/1, EP/R026645/1, EP/R027129/1, and EP/M013219/1. E.J. thanks the Swedish Research Council for their financial support in the form of an International Postdoc fellowship and Professor Clare Grey for the use of the Odyssey cluster for DFT calculations. The authors thank Mark Guyler (University of Nottingham) for the help with the magnetron sputtering laboratory set-up and Dr Michael Fay and Dr Emily Smith (University of Nottingham) for the helpful scientific discussions.

## Author contributions

I.C., A.W. and C.M. performed part of the ionic liquid synthesis and catalytic experiments. R.W.L., J.Y. and Z.Y.L. conducted TEM and AC-STEM measurements. J.P. performed the magnetron sputtering deposition of Pd species in ionic liquid and ICP-OES measurements. A.S. performed part of the ionic liquid synthesis and XPS measurements. G.A.R. conducted the Raman spectroscopy measurements. E.H.Å. conducted the classical molecular dynamics simulations with the PARCAS code. E.J. performed the DFT calculations. P.L., A.N.K. and J.A.F. designed the study, analysed the data and co-wrote the paper. All the authors discussed the results and commented on the manuscript.

## Competing interests

The authors declare no competing interests.
