## [Peer Review File · Nature Communications]

Title: Blurring the Boundary between Homogenous and Heterogeneous Catalysis using Palladium Nanoclusters with Dynamic SurfacesReviewers' comments:

Reviewer #1 (Remarks to the Author):

The authors have synthesized clusters of various sizes in ionic liquids by magnetron sputtering and report their structures and catalytic properties. In particular, it is claimed that small-sized Pd clusters have a "dynamic" structure, which manifests the characteristic catalytic properties attributed to them. However, the reviewers do not agree with the publication of this article, as it cannot be logically correct for the reasons stated below.

The dynamical cluster structure observed by STEM is considered to be driven by forces such as elastic or inelastic scattering of atoms by electron beam irradiation, instantaneous ionization associated with inelastic scattering, and local temperature increase, as previously reported[1]. It is clearly different from the conditions of the catalytic reaction, and it is dangerous to make short-sighted claims of relevance.

[1] Knez, D. et al. Modelling electron beam induced dynamics in metallic nanoclusters. *Ultramicroscopy* 192, 69–79 (2018).
10.1016/j.ultramic.2018.05.007

In Figure S8, there are single atoms around the cluster, which are thought to have been separated by electron irradiation. Similar atoms may be observed after the time-lapse in Figure S9 and S10. However, it is difficult to determine because the STEM image is, unfortunately, defocused. Figures S8 and S9 show changes over time, but they do not indicate how much time has passed. Overall, the dynamic structure is only an impression of the appearance.

The authors also claimed a dynamic structure based on the XAFS results. However, the validity cannot be judged because the increase in Pd-Pd bond distance with decreasing particle size is also common in Pd nanoparticles, even at larger sizes[2]. As previously reported, various possibilities, such as compound formation, should also be considered. In addition to the Fourier-transformed XAFS, more detailed structural analysis and discussion by curve-fitting analysis are required.

[2] Ohba, T. et al. An Origin for Lattice Expansion in PVP-Protected Small Pd Metal Nanoparticles. *Bull. Chem. Soc. Jpn.* 90, 720–727 (2017). 10.1246/bcsj.20170051

The smaller clusters showed higher catalytic activity and toxicity resistance. However, it is more reasonable to assume that the atomic Pd dissolved in the solution is the active species of the catalyst, as described in the previous report[3]. If the authors are to argue for the expression of activity by dynamic structure, it is necessary to deny the catalytic action by atomic Pd.

[3] Deraedt, C. & Astruc, D. "Homeopathic" palladium nanoparticle catalysis of cross carbon-carbon coupling reactions. *Acc. Chem. Res.* 47, 494–503 (2014). 10.1021/ar400168s

Reviewer #2 (Remarks to the Author):

This paper reports interesting results on small Pd nanoparticles sputtered in ion liquids. The authors claim that they find “nanoclusters with atomically dynamic surfaces” and these would be both acting as homogeneous and heterogeneous catalysts. Many results from both high-end experiments and calculation/simulations are presented. I cannot comment on the catalytic part and the simulations. However, many claims in the paper are not fully supported by the presented results. Therefore, I think that the paper needs a really thorough major revision. I suggest that the authors have to do further work to present fully convincing results to support their interpretations of the results, especially on the atomic scale structures and dynamics. There are several major issues which need to be revised to improve the quality of the manuscript. The claimed “atomic-scale investigations” are not visible, at least for TEM: the image quality provided in TEM images is not at atomic scale, rather nanoscale as the images are blurred (probably due to the ionic liquid).

Furthermore, the literature on sputtered metallic nanoparticles is not sufficiently considered. Especially, in the last years (2018-2020) there were many publications on sputtered nanoparticles also for catalytic applications and studied by TEM, also in atomic-scale resolution, which should be mentioned and discussed in this paper. Other authors reported optimized TEM sample preparation methods which yield high quality images of sputtered nanoparticles in ionic liquid. Authors should also provide FFT images of the single nanoparticles to show information on the crystallinity of the particles. It is unclear how the size histograms have been gathered. How many clusters/nanoparticles have been evaluated? For me the TEM images have a too-bad quality to be able to do this analysis properly.

Another important point is: the "dynamics" are not defined at all. In Fig. 1b it reads “images acquired over time”. I could not find any information on which time? Milliseconds, seconds, hours? The “dynamic character” is for me not deducible from these images. Furthermore: What about electron beam sample interactions?

Fig.1: The meaning and significance of the intensity histograms are unclear.

Some further issues:

The expressions “endangered elements”, “naked surfaces” etc. “hotly debated” sound strange to me and should be replaced by more scientific terms. The first part of the introduction should be re-written with more scientific content. What is meant by “highly selective growth” in the abstract, ...

Instead of Torr, A, and psi, SI units should be used.

Page 4: “a Pd core of ca. 1.8 nm”: values should be given as $x \text{ nm} \pm 0.x \text{ nm}$, there is no real meaning to “ca.”

Furthermore, core and shell are also not discernible. The time series is not defined.

How is a “significantly lower dynamics” defined. I don’t see any time information.

What about stability? How do the clusters look after catalytic experiment? Could the authors provide TEM images?

Reviewer #3 (Remarks to the Author):

The authors deposited metal clusters/particles into the selected ionic liquid by using a magnetron sputtering approach. The authors claimed that a diffuse dynamic shell of Pd atoms surrounding a densely populated Pd core is responsible for the observed activity of alkene cyclopropanation. The authors further stated that the dynamic Pd nanoclusters are resistant to mercury poisoning while compact Pd nanoclusters do not possess such property.

The use of ILs to support metal atoms/clusters/particles is not new. A recent publication (Chem 5, 2019, 3207) clearly demonstrated that ILs can stabilize metal atoms onto support surfaces. Several technical/scientific issues should be addressed prior to the publication of this manuscript.

1) Figure 1b clearly shows the presence of single Pd atoms within the IL. It is assumed that many such Pd atoms should exist in the IL, especially when the Pd loading level is low. Table S5 shows that the TON for extremely low loading levels of Pd is high. Therefore, Pd single atoms may have played a major role for the observed performance. From this perspective, it is misleading to claim that the diffuse shell of the Pd nanoclusters plays the most important role. In fact, one can consider some of these diffuse shells of Pd to be single atoms. If this is the case, then it’s not necessarily the dynamic nature of the Pd clusters that control the catalytic performance. Fig. 3b could be interpreted in terms of the amount of single Pd atoms.

2) Fig. S4-S6. For Pd particles with sizes $< 2 \text{ nm}$, one should not use the phase-contrast HRTEM/TEM to determine the particle sizes. One should use the aberration-corrected HAADF method to provide the particle size distributions. It does not make sense at all to count particles with sizes $< 1 \text{ nm}$ in images such as Fig. S4a. These samples may contain many Pd monomers and multimers!

3) Fig. S7. The image resolution of these HAADF images is poor. The single Pd atoms are not visible. Were these images obtained on a non-aberration-corrected STEM? Fig. S7b is clearly out of focus!

4) Fig. 1b and S8. The dynamic nature of these Pd clusters could be due to electron beam effects. Would the electron beam irradiation change the IL molecular structure? Even though these Pd clusters may

become dynamic in ILs one should not use HAADF images to confirm this unless careful investigation on the electron beam effects have been conducted. These images also show the presence of many single Pd atoms within the ILs.

Dear Reviewers,

Please find enclosed the revised version of the manuscript NCOMMS-20-11090 titled "*Blurring the Boundary between Homogenous and Heterogeneous Catalysts: Palladium Nanoclusters with Atomically Dynamic Surfaces*". We would like to thank all the reviewers for their valuable comments and suggestions. We have successfully addressing all critical points raised by the reviewers through a comprehensive revision of the manuscript, including the significantly extended ESI section. We believe that the revised manuscript is significantly improved.

Below, we enclose a point-by-point response to the reviewers' questions, and attach a revised version of the manuscript and ESI. The additions/revisions are highlighted in yellow in the main text and support information.

Reviewer #1 (Remarks to the Author):

The authors have synthesized clusters of various sizes in ionic liquids by magnetron sputtering and report their structures and catalytic properties. In particular, it is claimed that small-sized Pd clusters have a "dynamic" structure, which manifests the characteristic catalytic properties attributed to them. However, the reviewers do not agree with the publication of this article, as it cannot be logically correct for the reasons stated below.

1. The dynamical cluster structure observed by STEM is considered to be driven by forces such as elastic or inelastic scattering of atoms by electron beam irradiation, instantaneous ionization associated with inelastic scattering, and local temperature increase, as previously reported [1]. It is clearly different from the conditions of the catalytic reaction, and it is dangerous to make short-sighted claims of relevance.

[1] Knez, D. et al. Modelling electron beam induced dynamics in metallic nanoclusters. *Ultramicroscopy* 192, 69–79 (2018). 10.1016/j.ultramic.2018.05.007

We would like to thank the reviewer for highlighting this potential issue. Indeed, when used incorrectly, electron beam of TEM or STEM can be an invasive tool, changing the structure of observed materials (we have extensive expertise in the field of e-beam damage in nanomaterials i.e. <https://pubs.acs.org/doi/abs/10.1021/acs.accounts.7b00078>). We have been extremely careful not to induce any dynamics in the clusters by e-beam in this study, and we have clear evidence that there is no impact of e-beam on the materials in this investigation. The additional data in (Figure S6-7) proves this point.

On the second point, we carried out all our TEM and STEM imaging for Pd nanoclusters embedded in ionic liquids, same as those used for the reactions. It is known that ionic liquids remain in the same state in the vacuum of TEM/STEM as under normal conditions due to their extremely low vapor pressure. This allows us to approach the imaging conditions in TEM/STEM as close to realistic reactions conditions as possible without loss

of the atomic resolution. Furthermore, we have evaluated the physicochemical properties of Pd clusters not only by STEM/TEM but by using an arsenal of characterization techniques (XPS, XANES, EXFAS, Raman and DFT calculation). All these characterization data is fully consistent with the catalytic properties observed in the reaction kinetics and catalyst poisoning tests.

Therefore, this gives us confidence to correlate structure and catalytic properties of Pd nanoclusters.

2. In Figure S8, there are single atoms around the cluster, which are thought to have been separated by electron irradiation. Similar atoms may be observed after the time-lapse in Figure S9 and S10. However, it is difficult to determine because the STEM image is, unfortunately, defocused. Figures S8 and S9 show changes over time, but they do not indicate how much time has passed. Overall, the dynamic structure is only an impression of the appearance.

We are confident that there is no e-beam effect on the nature of Pd nanoclusters observed in STEM or TEM. In order to prove this point, we carried out additional TEM experiments to completely rule out the hypotheses of single Pd atoms around the core Pd cluster being separated by electron beam irradiation. The dynamic cluster sample was exposed to electron beam irradiation for 3 min (Figure S6-7). The images demonstrate that single Pd atoms were not being separated from Pd clusters for the duration of the imaging and the single Pd atom shell around cluster is a genuine feature of the sample. Furthermore, no electron beam-induced size or morphological changes on Pd clusters were observed.

These results are completely in line with the mercury tests performed in the catalytic experiments section. For instance, if the dynamic clusters were formed during electron beam irradiation, they should not demonstrate any catalytic activity when mercury is added to the reaction mixture, synonymous to the compact cluster; however, contrarily, catalytic activity was observed. This confirmed the pseudo-homogeneous character of the dynamic clusters in catalysis.

In this work, we are not using only one piece of evidence to support our findings, but several characterization techniques and catalytic experiments; therefore, this potential of this work must be appreciated holistically, not by the individual elements from which it is comprised.

We have carried out additional measurements and included a series of new higher-quality images have been added into the ESI (Figure S4 to S13).

3. The authors also claimed a dynamic structure based on the XAFS results. However, the validity cannot be judged because the increase in Pd-Pd bond distance with decreasing particle size is also common in Pd nanoparticles, even at larger sizes [2]. As previously reported, various possibilities, such as compound formation, should also be considered. In addition to the Fourier-transformed XAFS, more detailed structural analysis and discussion by curve-fitting analysis are required.

[2] Ohba, T. et al. An Origin for Lattice Expansion in PVP-Protected Small Pd Metal Nanoparticles. Bull. Chem. Soc. Jpn. 90, 720–727 (2017). 10.1246/bcsj.20170051

We thank the reviewer for the suggested reference, as it completely supports, rather than undermines, our findings and data interpretation (REF. 29).

- In this paper the authors investigated Pd NPs of 3.8 ± 1.3 nm; 4.0 ± 1.4 nm, 5.3 ± 2 nm and Pd foil (pg. 723 – Table 1). Interestingly, the authors did not find an increase of Pd-Pd bond distance between Pd (4 nm), P (5.3 nm) and Pd Foil. However, they found a slight increase on Pd-Pd bond distance for Pd (3.8 nm) when compared with the others samples (pg. 723 – Table 1).
- The TEM images of Pd (4.0 nm) and Pd (5.3 nm) clearly showed crystalline Pd NPs, whereas for Pd (3.8 nm), diffuse Pd NPs were observed with no d-spacing (Page 722 – Fig. 1). The XRD confirmed a more amorphous character for the sample (3.8 nm) when compared with the others samples (Page 722 – Fig. 2).
- In summary the authors concluded that the increase of Pd-Pd distance is associated to a greater degree of disorder in the Pd (3.8 nm) (Page 723 – left column, line 1-3) due to PdC_n formation during the reduction process (Page 726, conclusions), which the authors could not confirm.

Therefore, the paper suggested by the reviewer is in complete agreement with our results, where we associated the Pd-Pd bond distance elongation to the disorder of Pd dynamic clusters. *“Additionally, for 5Pd@[NTf₂] a slight shift to higher Pd-Pd bond distance compared to 30Pd@[NTf₂] and Pd bulk was observed, which can be ascribed to the dynamic Pd nanocluster behavior of relative low Pd loading in IL (5Pd@[NTf₂])”*

Regarding the possible coordination Pd to other elements, we also suggested in our EXFS and DFT analysis:

“Extended X-ray absorption fine structure (EXAFS) analysis of 5Pd@[NTf₂] and 30Pd@[NTf₂] show a peak at ca. 2.5 Å associated with Pd-Pd bond same as in bulk, as well as another peak at ca. 1.7 Å that we assigned to Pd atoms interaction with [C₄C₁Im][NTf₂] (Figure 2C). Density functional theory (DFT) modelling confirmed that the Pd coordination occurs via N and O atoms from the [NTf₂] anion for the Pd₁₃[C₄C₁Im][NTf₂],

whereas for the Pd₁[C₄C₁Im][NTf₂] complex the coordination occurs via C-atoms and O-atoms from the cation and anion, respectively (Figure 2e)."

For instance, the peak at ca. 1.7 nm is well known to be associated to Pd-C/O/N bonds and the separation of each contribution is challenging and should not be over-interpreted, as we describe in the manuscript. The peak at 2.5 nm is related to the same Pd-Pd bond as Pd foil.

Contrary to the reviewer's assertion, we did not attempt curve-fitting or simulation of this system, as can be observed in Figure 2 (details in the ESI). The data was calibrated using a Pd foil as reference and then normalized using Athena software, which is well-established and standard to treat this type of data (Nature Chemistry 2020 doi:10.1038/s41557-020-0446-z).

Finally, we would like to reinforce that, we do not claim the presence of dynamic clusters only based on the XAFS measurements and, it is based on several characterization techniques, such as TEM, STEM, XPS, DFT and catalytic experiments.

4. The smaller clusters showed higher catalytic activity and toxicity resistance. However, it is more reasonable to assume that the atomic Pd dissolved in the solution is the active species of the catalyst, as described in the previous report [3]. If the authors are to argue for the expression of activity by dynamic structure, it is necessary to deny the catalytic action by atomic Pd.

[3] Deraedt, C. & Astruc, D. "Homeopathic" palladium nanoparticle catalysis of cross carbon-carbon coupling reactions. *Acc. Chem. Res.* 47, 494–503 (2014). 10.1021/ar400168s

Within the concept of the dynamic Pd cluster, the single-atom catalysis is mere consequence of the dynamic nature at the atomic level. The reviewer seems to be implying that the pseudo-homogeneous mechanism where metal nanoclusters dissolve in the liquid phase during the catalytic reaction. This is not the case under our reaction conditions. We have proven this point by performing an additional experiment where dynamic Pd nanoclusters were retrieved from the reaction mixture directly onto a TEM grid. Our imaging demonstrated that the dynamic clusters remain the same in size and structure within the error of the measurement (Figure S30).

In the second comment, the term of "catalytic toxicity resistance" appears to make no sense in the field of catalysis. If the reviewer is referring to the catalytic activity, we study catalyst order in the reactions. It is important to note that a higher catalyst order does not necessarily suggest a higher catalytic activity which is not a topic of this work. A catalyst order refers to the amount of catalytically active sites present in a single turnover.

Reviewer #2 (Remarks to the Author):

This paper reports interesting results on small Pd nanoparticles sputtered in ion liquids. The authors claim that they find "nanoclusters with atomically dynamic surfaces" and these would be both acting as homogeneous and heterogeneous catalysts. Many results from both high-end experiments and calculation/simulations are presented. I cannot comment on the catalytic part and the simulations. However, many claims in the paper are not fully supported by the presented results. Therefore, I think that the paper needs a really thorough major revision. I suggest that the authors have to do further work to present fully convincing results to support their interpretations of the results, especially on the atomic scale structures and dynamics. There are several major issues which need to be revised to improve the quality of the manuscript. The claimed "atomic-scale investigations" are not visible, at least for TEM: the image quality provided in TEM images is not at atomic scale, rather nanoscale as the images are blurred (probably due to the ionic liquid).

1. Furthermore, the literature on sputtered metallic nanoparticles is not sufficiently considered. Especially, in the last years (2018-2020) there were many publications on sputtered nanoparticles also for catalytic applications and studied by TEM, also in atomic-scale resolution, which should be mentioned and discussed in this paper.

We would like to thank the reviewer this comment. There is a vast literature on metal nanoclusters and nanoparticles made by magnetron sputtering. Also, there is a growing body of work on nanoparticles sputtered into ionic liquids or/and in solid supports. We have added the Refs 17-18 to previous cite key references (Refs14-16 and 19-20). However, there is no reports in the literature between 2018-2020, on sputtered nanoparticles in *bulk ionic liquids for catalysis*. We would be grateful if the reviewer could direct us to the specific work they are referring to so we can amend our manuscript accordingly.

Other authors reported optimized TEM sample preparation methods which yield high quality images of sputtered nanoparticles in ionic liquid.

We significantly improved our preparation methods, which are described in ESI now. Please, see the preparation methods and the new higher-quality images added in the ESI (Figure S4 to S13).

Authors should also provide FFT images of the single nanoparticles to show information on the crystallinity of the particles.

It is provided in the ESI (Figures S9, S10 and S12).

It is unclear how the size histograms have been gathered. How many clusters/nanoparticles have been evaluated? For me the TEM images have a too-bad quality to be able to do this analysis properly.

We have described it in the ESI (TEM measurements section) and new higher-quality images have been added (Figures S4-S13).

2. Another important point is: the "dynamics" are not defined at all. In Fig. 1b it reads "images acquired over time". I could not find any information on which time? Milliseconds, seconds, hours? The "dynamic character" is for me not deducible from these images. Furthermore: What about electron beam sample interactions?

We agree with the reviewer that some details were missing in the original manuscript. In the revised version the STEM/TEM experimental work is now better described in the ESI (TEM measurements section).

We carried out new TEM experiments to completely rule out the hypotheses of single Pd atoms around the core Pd cluster being separated by electron beam irradiation. The dynamic cluster sample was exposed to electron beam irradiation for 3 min (Figure S6-7). The images demonstrate that single Pd atoms were not being separated from Pd clusters for the duration of the imaging and the single Pd atom shell around cluster is a genuine feature of the sample. Furthermore, no electron beam-induced size or morphological changes on Pd clusters were observed.

These results are completely in line with the mercury tests performed in the catalytic experiments section. For instance, if the dynamic clusters were formed during electron beam irradiation, they should not demonstrate any catalytic activity when mercury is added to the reaction mixture, synonymous to the compact cluster; however, contrarily, catalytic activity was observed. This confirmed the pseudo-homogeneous character of the dynamic clusters in catalysis.

In this work, we are not using only one piece of evidence to support our findings, but several characterization techniques and catalytic experiments cross-correlated with each other.

Fig.1: The meaning and significance of the intensity histograms are unclear.

The intensity bar (a.u.) in the Figure 1 refers to STEM image contrast which correlates with the number of Pd atoms in the image. We have added an explanatory statement in the figure caption.

3. Some further issues: The expressions "endangered elements", "naked surfaces" etc. "hotly debated" sound strange to me and should be replaced by more scientific terms. The first part of the introduction should be re-written with more scientific content. What is meant by "highly selective growth" in the abstract, ...

We have modified the expressions suggested by reviewer, and replaced these phrases as follows: "endangered elements" with "rare elements", "naked surfaces" with "clean surfaces", "hotly debated" with "widely debated", "highly selective growth" with "size-controlled formation of nanoclusters".

4. Instead of Torr, A, and psi, SI units should be used.

The changes requested by reviewer were made in the main manuscript and ESI.

5. Page 4: "a Pd core of ca. 1.8 nm": values should be given as $x \text{ nm} \pm 0.x \text{ nm}$, there is no real meaning to "ca." Furthermore, core and shell are also not discernible. The time series is not defined. How is a "significantly lower dynamics" defined. I don't see any time information.

The modification requested by the reviewer in page 4 was made. The lower dynamics term was used for comparison between the Pd dynamic clusters and compact clusters specifically in this work. The "time" information is now given in the ESI (Figures S8, S11 and S13).

6. What about stability? How do the clusters look after catalytic experiment? Could the authors provide TEM images?

TEM measurements were performed where dynamic Pd nanoclusters were retrieved from the reaction mixture directly onto a TEM grid. Our imaging demonstrated that the dynamic clusters remain the same in size and structure within the error of the measurement (Figure S30).

Reviewer #3 (Remarks to the Author):

The authors deposited metal clusters/particles into the selected ionic liquid by using a magnetron sputtering approach. The authors claimed that a diffuse dynamic shell of Pd atoms surrounding a densely populated Pd core is responsible for the observed activity of alkene cyclopropanation.

This is not a correct interpretation of our results. We observe that all catalysts display catalytic activity. However, catalysts with a dynamic shell displayed properties typically found in homogenous catalysts. When no dynamic shell are present, the catalytic properties are closer to a heterogeneous catalyst (see mercury test and catalyst order experiments - Figure 3).

The authors further stated that the dynamic Pd nanoclusters are resistant to mercury poisoning while compact Pd nanoclusters do not possess such property.

1. The use of ILs to support metal atoms/clusters/particles is not new. A recent publication (Chem 5, 2019, 3207) clearly demonstrated that ILs can stabilize metal atoms onto support surfaces. Several technical/scientific issues should be addressed prior to the publication of this manuscript.

At the core of our approach is direct magnetron sputtering of metal nanoclusters into IL, in their highly active, native form, without any need of wet chemistry, which is different to the paper cited above and most of other works on metal nanocluster catalysis in IL. Our unique methodology to nanocluster fabrication allows investigation of the most fundamental features of nanocatalysts, such as their structure-property relationship at the atomic scale.

2. Figure 1b clearly shows the presence of single Pd atoms within the IL. It is assumed that many such Pd atoms should exist in the IL, especially when the Pd loading level is low. Table S5 shows that the TON for extremely low loading levels of Pd is high.

We did not change the Pd concentration in the catalytic experiments. The Pd loading for all catalytic experiment was 5 mol%.

Therefore, Pd single atoms may have played a major role for the observed performance. From this perspective, it is misleading to claim that the diffuse shell of the Pd nanoclusters plays the most important role. In fact, one can consider some of these diffuse shells of Pd to be single atoms. If this is the case, then it's not necessarily the dynamic nature of the Pd clusters that control the catalytic performance. Fig. 3b could be interpreted in terms of the amount of single Pd atoms.

Within the concept of the dynamic Pd cluster, the single-atom catalysis is mere consequence of the dynamic nature at the atomic level. The reviewer seems to be implying that the pseudo-homogeneous mechanism where metal nanoclusters dissolve in the liquid phase during the reaction. This is not the case under our reaction conditions. We have proven this point by performing an additional experiment where dynamic Pd nanoclusters were retrieved from the reaction mixture directly onto a TEM grid. Our imaging demonstrated that the dynamic clusters remain the same in size and structure within the error of the measurement (Figure S30).

In contrast, the "compact" clusters exhibit entirely heterogeneous behavior under the same conditions (catalyst order is lower than 1 and Hg poisoning tests).

3. Fig. S4-S6. For Pd particles with sizes < 2 nm, one should not use the phase-contrast HRTEM/TEM to determine the particle sizes. One should use the aberration-corrected HAADF method to provide the particle size distributions. It does not make sense at all to count particles with sizes < 1 nm in images such as Fig. S4a. These samples may contain many Pd monomers and multimers!

Our aberration-corrected STEM shows no monomers or multimers in the nanocatalysts made by our method, which is also corroborated by the Hg test (please see earlier comments). HRTEM imaging has been improved and extended, with a series of new high-quality images now added in the ESI in the revised manuscript (Figure S4 to S13). The additional data are in completely agreement with our previous conclusions.

4. Fig. S7. The image resolution of these HAADF images is poor. The single Pd atoms are not visible. Were these images obtained on a non-aberration-corrected STEM? Fig. S7b is clearly out of focus!

This figure in ESI has been replaced as requested (Figure S4 to S13).

5. Fig. 1b and S8. The dynamic nature of these Pd clusters could be due to electron beam effects. Would the electron beam irradiation change the IL molecular structure? Even though these Pd clusters may become dynamic in ILs one should not use HAADF images to confirm this unless careful investigation on the electron beam effects have been conducted. These images also show the presence of many single Pd atoms within the ILs.

We carried out new TEM experiments to completely rule out the hypotheses of single Pd atoms around the core Pd cluster being separated by electron beam irradiation. The dynamic cluster sample was exposed to electron beam irradiation for 3 min (Figure S6-7). The images demonstrate that single Pd atoms were not being separated from Pd clusters for the duration of the imaging and the single Pd atom shell around cluster is a genuine feature of the sample. Furthermore, no electron beam-induced size or morphological changes on Pd clusters were observed.

These results are completely in line with the mercury tests performed in the catalytic experiments section. For instance, if the dynamic clusters were formed during electron beam irradiation, they should not demonstrate any catalytic activity when mercury is added to the reaction mixture, synonymous to the compact cluster; however, contrarily, catalytic activity was observed. This confirmed the pseudo-homogeneous character of the dynamic clusters in catalysis.

In this work, we are not using only one piece of evidence to support our findings, but several characterization techniques and catalytic experiments; therefore, this potential of this work must be appreciated holistically, not by the individual elements from which it is comprised.

REVIEWER COMMENTS

Reviewer #4 (Remarks to the Author):

Combining the manuscript and the comments of the previously three reviewers, I think that three reviewers made valid points. Yes, the work is interesting. I agree that it provides an important insight in bridging homogeneous and heterogeneous catalysis. However, the "dynamics" structures seems to be rather dubious. The results are remarkable but not convincing. This work can not be published in Nature Communications in the current form. Here are some specific criticisms:

1. The "dynamics" cluster structure are not convincing. If possible, I know it is difficult, some in-situ characterizations have to be carried to determine "dynamics" cluster structures on the Pd@[C4C1Im][NTf2] catalyst. In addition, the role of "dynamics" structure on catalytic performance should be discussed in detail and further clarified.
2. Figure 1b and S8 show the presence of single Pd atoms within the IL. One should use the aberration-corrected HAADF to confirm whether there is the single atoms in Pd@[C4C1Im][NTf2] catalyst. Which one is active species, Pd cluster or single Pd atoms? Some control experiments should be performed to determine the possible active site on the Pd@[C4C1Im][NTf2] catalyst.
3. Reusability of the Pd@[C4C1Im][NTf2] should be performed. The TEM of reused three times samples should be provided, since it's very important to compare the differences of fresh and reused samples in the structure. In addition, during the cycle experiment of the reused catalysts, the loss of palladium should be considered. Therefore, the Pd loading of the reused Pd@[C4C1Im][NTf2] catalyst should be added. Additionally, the reaction solutions should be checked for traces of leached Pd with IPC-AES.
4. To determine the homogeneous or heterogeneous nature of the catalysis, the authors should perform a filtration experiment.

Reviewer #5 (Remarks to the Author):

After the revision, this manuscript might be accepted after the changes in the following:

Authors claimed that "The new Pd@[C4C1Im][NTf2] system is the first example where the features of homogeneous and heterogeneous are combined within the same material". In fact, many phase transfer catalysts containing metal nanoparticles have the feature of both homogeneous and heterogeneous catalysis. I think that authors should delete the words such as "first time".

Dear Reviewers,

Please find enclosed the revised version of the manuscript NCOMMS-20-11090B titled "*Blurring the Boundary between Homogenous and Heterogeneous Catalysts: Palladium Nanoclusters with Atomically Dynamic Surfaces*". We would like to thank all the reviewers for their valuable comments and suggestions. We have successfully addressed all the critical points raised by the reviewers through a comprehensive revision of the manuscript, including all additional experiments requested. We believe that the revised manuscript is significantly improved.

Below, we enclose a point-by-point response to the reviewers' questions, and attach a revised version of the manuscript and ESI. The additions/revisions are highlighted in yellow in the main text and supporting information.

Reviewer #4 (Remarks to the Author):

Comment: Combining the manuscript and the comments of the previously three reviewers, I think that three reviewers made valid points. Yes, the work is interesting. I agree that it provides an important insight in bridging homogeneous and heterogeneous catalysis. However, the "dynamics" structures seems to be rather dubious. The results are remarkable but not convincing. This work cannot be published in Nature Communications in the current form. Here are some specific criticisms:

Comment: The "dynamics" cluster structure are not convincing. If possible, I know it is difficult, some in-situ characterizations have to be carried to determine "dynamics" cluster structures on the Pd@[C4C1Im][NTf2] catalyst.

Response: The dynamic nature of the nanoclusters was probed by time-resolved HRTEM and AC-STEM imaging, performed directly in the solvent (ionic liquid) and the same temperature as the macroscopic reaction, which is the closest set of conditions to the real reaction possible for atomic scale imaging (Figures 1b, S7, S8). Specifically, the 5Pd@[NTf₂] system exhibited significant dynamics in the structure at the atomic level in individual nanoclusters, while the overall integrity of the nanoclusters was maintained, i.e. no leaching, coarsening or coalescence (Figures 1b, S7, S8). The observed in situ single-particle dynamics correlates well with the macroscopic observations of their catalytic performance (Figure 3, Table S10, S11).

Comment: In addition, the role of "dynamics" structure on catalytic performance should be discussed in detail and further clarified.

Reply: We have carried out a comprehensive set of experiments that conclusively demonstrates that dynamics of Pd nanocluster plays a key role in the catalytic performance. Specifically, we have performed selective poisoning tests using dibenzo[*a,e*]cyclooctene (DCT) which selectively binds to single sites [Angew. Chem. Int. Ed. 2014, 53, 3722–3726 (ESI); Adv. Synth. Catal. 2018, 360, 1833–1840]. This approach allowed to prove that single Pd atoms are not the active species, and confirmed that dynamic Pd nanoclusters are indeed the active sites for this reaction (Table S11). Furthermore, catalytic system 5Pd@NTf₂ does not show activity after a filtration test (Table S10), which reinforces this concept of dynamic clusters as active catalysts.

Comment: Figure 1b and S8 show the presence of single Pd atoms within the IL. One should use the aberration-corrected HAADF to confirm whether there is the single atoms in Pd@[C4C1Im][NTf2] catalyst.

Response: Our aberration-corrected STEM imaging demonstrates that the nanocatalysts made by our method exist in a nanocluster form in solvent (Figures 1b, S7, S8), and the absence of single Pd atoms is also corroborated by Hg test (Figure 1b), DCT test (Table S11) and filtration experiments (Table S10). A combination of the atomic-scale imaging with these bulk scale analyses of the catalyst in action provide a substantial evidence that single Pd atom catalysis plays no role in this reaction.

Comment: Which one is active species, Pd cluster or single Pd atoms? Some control experiments should be performed to determine the possible active site on the Pd@[C4C1Im][NTf2] catalyst.

Response: this question is answered in an earlier response (please see above).

Comment: Reusability of the Pd@[C4C1Im][NTf2] should be performed. The TEM of reused three times samples should be provided, since it's very important to compare the differences of fresh and reused samples in the structure. In addition, during the cycle experiment of the reused catalysts, the loss of palladium should be considered. Therefore, the Pd loading of the reused Pd@[C4C1Im][NTf2] catalyst should be added. Additionally, the reaction solutions should be checked for traces of leached Pd with IPC-AES.

Response: As requested, the recycling experiments and analysis of the reaction mixture by ICP-OES have been added to the manuscript. 5Pd@NTf₂ have been shown to be recyclable and maintain its activity for 3 cycles (Table S12).

Furthermore, TEM experiment where dynamic Pd nanoclusters were retrieved from the reaction mixture directly onto a TEM grid demonstrated that the dynamic clusters remain unchanged in size and structure after the reaction (Figure S30).

Comment: To determine the homogeneous or heterogeneous nature of the catalysis, the authors should perform a filtration experiment.

Response: As requested, 5Pd@NTf₂ has been subjected to a filtration test (Table S10), which revealed no activity after nanocluster filtration, thus proving that the dynamic nanocluster are the active sites.

Reviewer #5 (Remarks to the Author):

Comment: After the revision, this manuscript might be accepted after the changes in the following:

Authors claimed that "The new Pd@[C4C1Im][NTf2] system is the first example where the features of homogeneous and heterogeneous are combined within the same material". In fact, many phase transfer catalysts containing metal nanoparticles have the feature of both homogeneous and heterogeneous catalysis. I think that authors should delete the words such as "first time".

Response: We would like to thank the reviewer for the comment. As requested, we have removed the expressions suggested by reviewer.

REVIEWERS' COMMENTS

Reviewer #4 (Remarks to the Author):

The authors answered the remarks properly and I have no further comments to do.